# Biology and Rearing of an Emerging Sugar Beet Pest: The Planthopper *Pentastiridius leporinus*

**DOI:** 10.3390/insects13070656

**Published:** 2022-07-21

**Authors:** Sarah Christin Behrmann, Natasha Witczak, Christian Lang, Manuela Schieler, Anna Dettweiler, Benno Kleinhenz, Mareike Schwind, Andreas Vilcinskas, Kwang-Zin Lee

**Affiliations:** 1Department of Insect Biotechnology, Justus Liebig University of Giessen, Heinrich-Buff-Ring 26, D-35392 Giessen, Germany; sarah-christine.behrmann@ag.uni-giessen.de (S.C.B.); natasha.witczak@umwelt.uni-giessen.de (N.W.); andreas.vilcinskas@agrar.uni-giessen.de (A.V.); 2Fraunhofer Institute for Molecular Biology and Applied Ecology, Ohlebergsweg 12, D-35394 Giessen, Germany; 3Association of Hessian-Palatinate Sugar Beet Growers e.V., Rathenaustraße 10, D-67547 Worms, Germany; lang@ruebe.info (C.L.); dettweiler@ruebe.info (A.D.); schwind@ruebe.info (M.S.); 4Central Institute for Decision Support Systems in Crop Protection (ZEPP), Rüdesheimer Straße 60, D-55545 Bad Kreuznach, Germany; manuela.schieler@dlr.rlp.de (M.S.); benno.kleinhenz@dlr.rlp.de (B.K.)

**Keywords:** SBR, *Pentastiridius leporinus*, sugar beet, Cixiidae, rearing, monitoring, *Beta vulgaris*, stolbur phytoplasma, proteobacteria

## Abstract

**Simple Summary:**

Syndrome des Basses Richesses (SBR) is a bacterial yellowing disease of sugar beet (*Beta vulgaris*) that reduces the sugar content of the root tissue significantly. There is currently neither a cure for the disease nor effective control measures for the main insect vector, the Cixiid planthopper *Pentastiridius leporinus*. SBR is therefore spreading in Germany and has caused substantial yield losses in recent years. The development of effective control measures requires a better understanding of *P. leporinus* biology, including its life cycle and feeding behavior. We therefore established methods for the continuous mass rearing of *P. leporinus*. In host plant choice experiments, we found that the nymphs are polyphagous when offered sugar beet, winter wheat or maize. Life cycle data indicated that adult planthoppers emerge 170 days after hatching, having passed through five larval instar stages. We developed a state-of-the-art qRT-PCR protocol using TaqMan probes to study the prevalence of two bona fide SBR pathogens: *Candidatus* Arsenophonus phytopathogenicus (Gammaproteobacteria) and *Candidatus* Phytoplasma solani (stolbur phytoplasma). These were identified during field studies in newly-infected regions of Rhineland-Palatinate and south Hesse, Germany, where we also observed insect mobility patterns in two consecutive years, including the abundance of adults at four locations and soil-depth monitoring of nymphs. We documented the spread of *P. leporinus* northward and eastward in Germany, accompanied by an increase in the number of SBR-carrying planthoppers. Interestingly, *P. leporinus* does not appear to hibernate during winter. Stolbur phytoplasma has a significant impact on SBR pathology in sugar beet.

**Abstract:**

The rapid spread of the bacterial yellowing disease Syndrome des Basses Richesses (SBR) has a major impact on sugar beet (*Beta vulgaris*) cultivation in Germany, resulting in significant yield losses. SBR-causing bacteria are transmitted by insects, mainly the Cixiid planthopper *Pentastiridius leporinus*. However, little is known about the biology of this emerging vector, including its life cycle, oviposition, developmental stages, diapauses, and feeding behavior. Continuous mass rearing is required for the comprehensive analysis of this insect. Here we describe the development of mass rearing techniques for *P. leporinus*, allowing us to investigate life cycle and ecological traits, such as host plant choice, in order to design agronomic measures that can interrupt the life cycle of nymphs in the soil. We also conducted field studies in recently-infected regions of Rhineland-Palatinate and south Hesse, Germany, to study insect mobility patterns and abundance at four locations during two consecutive years. The soil-depth monitoring of nymphs revealed the movement of the instars through different soil layers. Finally, we determined the prevalence of SBR-causing bacteria by designing TaqMan probes specific for two *bona fide* SBR pathogens: *Candidatus* Arsenophonus phytopathogenicus (Gammaproteobacteria) and *Candidatus* Phytoplasma solani (stolbur phytoplasma). Our data suggest that *P. leporinus* is spreading northward and eastward in Germany, additionally, the abundance of SBR-carrying planthoppers is increasing. Interestingly, *P. leporinus* does not appear to hibernate during winter, and is polyphagous as a nymph. Stolbur phytoplasma has a significant impact on SBR pathology in sugar beet.

## 1. Introduction

The planthopper *Pentastiridius leporinus* belongs to the family Cixiidae in the infraorder Fulgoromorpha. Cixiid species are small hemipterans that feed as adult stages on aboveground plant material, whereas their nymph stages feed below the ground on root sap. Many cixiid species are economically important because they transmit bacterial pathogens to plants. *P. leporinus* is a vector of the bacterial yellowing disease Syndrome des Basses Richesses (SBR) in sugar beet (*Beta vulgaris*), and large populations of this insect are found in sugar beet crops [1]. The symptoms of SBR include yellowing of older leaves, lancet-shaped leaf deformations, and necrosis of the vascular tissue, which significantly reduces the sugar content of root tissue and results in lower yields. SBR was first identified in France [2,3] but is now spreading rapidly in southern and eastern Germany, threatening the profitability of the sugar beet industry [4].

In central Europe, *P. leporinus* produces one generation per year. After mating at the end of June, beginning of July, the females deposit their egg mass in the soil and the nymphs continue to develop in the topsoil. Adult females and nymphs produce filamentous wax structures that protect them and the eggs from predation [5]. Sugar beet is harvested in fall, and winter wheat is sown shortly afterwards, providing the *P. leporinus* nymphs with an overwinter/spring food source until they emerge as adults and seek a new sugar beet crop [6].

SBR is associated with two different bacterial pathogens, *Candidatus* Arsenophonus phytopathogenicus and *Candidatus* Phytoplasma solani, both of which are transmitted by insect vectors and limited to the phloem [7,8]. *Candidatus* Arsenophonus phytopathogenicus (phylum Gammaproteobacteria), hereafter abbreviated to SBRars, is an intracellular bacterium that is transmitted vertically from *P. leporinus* females to their offspring with an efficiency of ~30%, and with a lower efficiency by males due to the presence of fewer bacteria in the male salivary gland [9]. *Candidatus* Phytoplasma solani, hereafter abbreviated to SBRps, is commonly known as stolbur phytoplasma. Phytoplasmas are bacterial pathogens (family *Acholeplasmataceae*) that lack a cell wall and colonize the phloem sieve tubes of plants, resulting in various economically important crop diseases [10]. In Serbia, stolbur phytoplasma is also associated with a different disease of sugar beet, known as rubbery tap root disease [11].

There are currently no effective measures against either of the SBR pathogens or the insect vector. Effective vector control requires a comprehensive knowledge of the distribution and behavior of *P. leporinus*, but the lack of mass breeding protocols makes it difficult to assess life cycle and life history traits in order to model the spread of the disease and test biological and environmentally safe alternative control measures in the laboratory and greenhouse. We therefore set out to develop the first mass rearing methods for this insect species. Furthermore, we gathered information on the flight behavior of the planthopper in different regions to determine the most suitable application period for any new control measures and investigated the subterranean movement of nymphs as a function of soil temperature in order to design strategies that interrupt the *P. leporinus* life cycle in the soil. Finally, we screened for both SBR pathogens at four sites in Rhineland-Palatinate and Hesse to monitor the spread of the disease in Germany. A better understanding of *P. leporinus* biology and the SBR pathogens will facilitate the development of environmentally friendly control strategies and assist with pest management decisions to control the spread of SBR.

## 2. Materials and Methods

### 2.1. Field Collection of Adults

We collected *P. leporinus* adults from commercial sugar beet fields during July 2020 in Biblis (49°41′52.6″ N 8°27′12.8″ E) and during June and July 2021 in Bickenbach (49°45′46.0″ N 8°33′36.2″ E). Both fields are located in the Rhine Valley close to the border of the Odenwald (89 m above sea level) in Hesse. Adults were collected using a sweep net and were immediately transferred to the greenhouse for rearing.

### 2.2. Rearing

Field-collected *P. leporinus* adults were placed directly on 8-week-old sugar beet plants (KWS Annarosa) growing in Göttinger 3-L pots (Lamprecht-Verpackungen, Göttingen, Germany) filled with Frühstorfer potting soil LD 80 (Heinrichs, Ingelheim, Germany) and covered with expanded clay (Floragard, Oldenburg, Germany). The insects were kept in groups of ~400 in a Cavea PopUp size M net cage (Howitec, Bolsward, Netherlands). The sugar beet plant in the cage was changed every 7 days. Plants were watered moderately twice weekly. The temperature in the greenhouse chamber was set to 22/16 °C day/night with a 16 h photoperiod.

In 2020, egg clusters were left inside the pot, allowing the nymphs to develop on the sugar beet roots. When the plant wilted (due to SBR infection), the nymphs were transferred to a new sugar beet plant. In 2021, sugar beet plants from the adult cages were carefully screened for eggs, which were transferred to a climate chamber (continuous 22 °C and 60% relative humidity, 16 h photoperiod). Mass rearing was carried out in GODMORGEN plastic boxes (IKEA Deutschland, München, Germany) filled with the granular soil additive GeoHumus (GeoHumus, Frankfurt/Main, Germany) to maintain humidity. Young sugar beet plants were placed in the boxes as a food source and were replaced weekly.

### 2.3. Life Cycle Data

Solitary nymphs were maintained in 60 mM Petri dishes with GeoHumus substrate and 1 × 1 × 1 cm cubes of young sugar beet roots as nutrition. The developmental stages were documented and measured using a VHX-5000 digital microscope (Keyence Deutschland, Neu-Isenburg, Germany) on a weekly basis for 34 weeks. The body size of individual nymphs was measured along the lateral axis [12] and the pigmentation of the nymphs was used for further characterization.

### 2.4. Host Plant Choice Experiments

Sugar beet, wheat and maize were presented to *P. leporinus* nymphs (the first progeny of field-collected adults) to determine their host plant preference. The host choice experiments were carried out using five replicate cohorts of 30 nymphs. The number of nymphs on each host plant was assessed after 1, 3, 6, 12, 24 and 48 h. The experimental setup consisted of four areas; a central area created by a plastic container (8 × 10.5 × 9 cm) from which three transparent plastic pipes (1.7 cm diameter) led to three separate plastic containers holding the different plants. The nymphs were released to the central area. The stalks of the plants were wrapped with 2 cm foam strips and fixed in the perforated lids of the containers, to make the roots available for the nymphs.

The host plants were grown in a hydroponic system. Maize and wheat seeds were transferred to a sandwich culture [13] and grown in 1 mM CaSO_4_, whereas sugar beet seed was grown in 1 mM CaSO_4_ containing 10 µM boric acid. Before germination, all cultures were coated with the seed treatment Maxim XL (Syngenta, Basel, Switzerland) containing the fungicides mefenoxam and fludioxonil. Sandwich cultures were grown under controlled conditions from day 4 (26/18 °C day/night, 16-h photoperiod). On day 8, the seedlings were transferred to 2.5-L containers filled with semi-concentrated nutrient solution containing the following nutrient salts: calcium nitrate tetrahydrate, potassium sulfate, magnesium sulfate heptahydrate, calcium chloride dihydrate, monobasic potassium phosphate, ferric EDTA, boric acid, manganese(II) sulfate monohydrate, zinc sulfate solution, copper(II) sulfate pentahydrate, ammonium heptamolybdate tetrahydrate and nickel(II) sulfate hexahydrate [14]. The containers were maintained under controlled conditions for a further 4 days (21/16 °C day/night, 16 h photoperiod). The stalks of the seedlings were then wrapped with 2 cm foam strips and fixed in the perforated lids of the containers, into which aeration tubes were inserted. An Osaga LK 60-piston compressor (Osaga, Buchholz, Germany) was used to circulate air in the nutrient solutions. On day 12, the plants were refreshed with fully-concentrated nutrient solution, which was changed weekly. The host choice experiments were initiated when the plants reached BBCH stage 15 (sugar beet and maize) or 31 (wheat).

### 2.5. Abundance of Adults in the Field

To measure the abundance of *P. leporinus*, 10 × 25 cm yellow sticky traps (Aeroxon, Waiblingen, Germany) were set up at four locations in Rhineland-Palatinate and south Hesse in the summer of 2020 and 2021. The sites were commercial sugar beet fields in Riedstadt (2020, 49°51′20.34″ N 8°23′47.67″ E; 2021, 49°52′18.2″ N 8°24′53.1″ E), Bickenbach (2020, 49°46′04.4″ N 8°33′30.9″ E; 2021, 49°46′03.2″ N 8°33′33.0″ E), Monsheim (2020, 49°37′59.2″ N 8°14′02.0″ E; 2021, 49°38′15.1″ N 8°14′42.9″ E) and Steinweiler (2020, 49°07′07.0″ N 8°09′03.6″ E; 2021, 49°06′29.8″ N 8°09′29.6″ E). Three traps were set up per site at a distance of 10, 30 and 50 m from the edge of the field on a pole at a height of 10 cm above the sugar beet vegetation. The panels were set up in the first week of May and the sampling was finished after two continuous weeks without catches. The sticky traps were replaced weekly and the number of adult *P. leporinus* on each trap was determined.

### 2.6. Subterranean Nymph Movement

From mid-November 2020 until June 2021, we excavated soil from winter wheat fields following a heavily SBR-infested sugar beet crop (2020, 49°37′06.4″ N 8°07′45.5″ E). After the adult planthoppers had left the field in June 2021, the study was continued in a nearby sugar beet field (2021, 49°37′51.2″ N 8°07′00.3″ E) until March 2022, where peas were sown in spring after the sugar beet harvest. Once a week, six soil samples (25 × 25 × 25 cm) were taken at marked points to determine the whereabouts of nymphs during the course of the year. Each soil sample was sliced in 5 cm increments and the number of *P. leporinus* nymphs per slice was counted. After each survey, the sampling points were moved 1 m in different directions on six sampling lines. The study was conducted for 66 weeks in total. The soil temperature was recorded using a Tinytag Plus 2 TPG-4017 (Gemini Data Loggers, Chichester, UK) at 5–10 cm depth.

### 2.7. Detection of SBRars and SBRps by qRT-PCR

Trapped *P. leporinus* specimens were tested for infection with SBRars and SBRps in 2020 and 2021 by examining 343 and 296 planthoppers, respectively. In the flight monitoring, up to 10 planthoppers per week collected from four sites were transferred to Eppendorf tubes containing 70% ethanol, and DNA was extracted using the Blood&Tissue Kit (Qiagen, Hilden, Germany). The detection of SBRars and SBRps was determined by qRT-PCR (Applied Biosystems, Foster City, CA, USA). For SBRars, we used primers SBRars_qRTforKL437 (5′-CAC AAG CGG TGG AGC ATG TG-3′) and SBRars_qRTrevKL438 (5′-ACT CCC ATG GTG TGA CGG-3′) with the FAM/TAMRA-labeled probe SBRars_qRTprobeKL457 (5′-FAM-TCT CAA TTA CTC ATA GTG AGA T-TamRa-3′). For SBRps, we used primers SBRps_qRTforKL464 (5′-TGG AGG TTA TCA GAA GCA CAG-3′) and SBRps_qRTrevKL465 (5′-TGC TAA AGT CCC CAA CTT AAT G-3′) with the FAM/TAMRA-labeled probe SBRps_qRTprobeKL466 (5′-FAM-ATG TTG GGT TAA GTC CCG CAA CGA GCG CAA CC-TamRa-3′). Targets were amplified using the Luna Universal Probe qPCR master mix kit (New England Biolabs, Ipswich, MA, USA). The total DNA amount of the samples was adjusted to 10 ng/µL. A plasmid carrying the SBRars and SBRps target was used to prepare a standard curve including 10-fold dilution series. The samples were heated to 55 °C for 10 min, then 95 °C for 1 min, followed by 40 cycles of 95 °C for 10 s and 60 °C for 30 s in the amplification and fluorescence detection steps, respectively. The bacterial load was calculated against the standard curve using ABI 7500 System SDS Software (Applied Biosystems). In brief, we considered a threshold of 10 copies for SBRars and a threshold of 3 copies for SBRps, or for both cases a CT value higher than 35 as negative results.

### 2.8. Statistical Analysis

For growth rate analysis, a linear model was fitted to the lateral length data plotted against the number of days after hatching. Molting events were determined by observing the shed cuticle remnants, called exuviae, and were assigned to one of five developmental transitions based on the characteristic physical attributes of the nymphs. Host plant choice data were analyzed in GraphPad Prism v9.1.2 for Windows (GraphPad Software, San Diego, CA, USA) using Friedman’s test followed by Dunn’s multiple comparisons test. Subterranean nymph movement data were assessed by one-way analysis of variance (ANOVA) followed by Bonferroni’s multiple comparisons test, also in GraphPad Prism. Adult abundance data were analyzed in R v4.1.2 (R Core Team, Vienna, Austria, 2021) using a Mann–Whitney U-test and a Kruskal–Wallis test for data with non-normal distribution. 

## 3. Results

### 3.1. Successful Mass Rearing of P. leporinus Facilitates Life Cycle Data Analysis

The rearing of *P. leporinus* was successful in two different settings. In 2020, we observed oviposition by caught adults (P0) and the development of nymphs in a climate chamber. The second generation of adults (F1) started molting after three months under climate-controlled conditions in August 2020 and continued emerging until January 2021. During this period, 905 adults emerged and deposited eggs (F2). Nymphs from the second generation hatched in sugar beet pots and were transferred to the climate chamber for use in other projects.

From the initial *P. leporinus* population (collected in the field in summer 2021) we studied two successive generations under climate chamber conditions (Figure 1). Under climate-controlled conditions, a growth curve with molting events was prepared based on the data from 60 individuals over 34 weeks (Figure 2). The mean molting time points (M1–M5, dark gray vertical lines) were at 32, 44, 77, 90 and 170 days after hatching, respectively. The mean size of first instar was 1053 ± 30 µm, increasing to 1600 ± 34 µm for the second instar, 2300 ± 60 µm for the third, 3320 ± 83 µm for the fourth, and 4270 ± 43 µm for the fifth. The mean size of adult specimens was 5028 ± 372 µm. Waxy filaments were produced by all five instars. We observed no morphological differences between male and female specimens at any nymph developmental stage.

### 3.2. P. leporinus Nymphs Are Polyphagous with Weak Host Preferences

The ability of *P. leporinus* to survive for several months in the absence of a sugar beet crop suggests that the nymphs feed on alternate hosts. We therefore carried out host choice tests in which nymphs were offered sugar beet and two typical crops planted after the sugar beet harvest: winter wheat and maize. We found that significantly greater numbers of nymphs were attracted to sugar beet and maize than to wheat and the central area, but sugar beet and maize were equally attractive (Figure 3).

### 3.3. Subterranean P. leporinus Nymphs Show Temperature-Dependent Movement

Field studies revealed that nymphs are present in the soil throughout the year, and they do not hibernate over winter. However, the movement of individual nymphs is limited by the colder temperatures. In 2021, the first oviposition was observed on July 6th in Bubenheim (49°37′51.2″ N 8°07′00.3″ E). From November 2020 to March 2022, 488 individuals were found. The number of nymphs per sampling point and soil layer varied between 1 and 36, with an average of 2.6 ± 3.94. Nymphs were found in the topsoil layer (0–10 cm) at a mean temperature of 8.6 °C. When the temperature in the topsoil dropped below 5.6 °C, nymph abundance increased significantly in the next deeper layer (10–20 cm) and in the subsoil (20–30 cm) (Figure 4).

### 3.4. Abundance of Adults of P. leporinus and Prevalence of SBR Pathogens

In 2020, we caught 800 *P. leporinus* individuals at all sites using yellow sticky traps, 95% of which were males. In 2021, the number of trapped individuals was 2022. The longest flight activity was tracked in Bickenbach. The 2020 flight period lasted from calendar week 21 (mid-May) until calendar week 38 (late September). In 2021, the flight period in Bickenbach was shorter, from calendar week 23 (mid-June) until calendar week 33 (mid-August). In order to compare the abundance of *P. leporinus* between the years, we defined the main flight period starting from calendar week 23 and lasting until week 31, when planthoppers were present at all sites. In Bickenbach, Steinweiler and Monsheim a significantly greater number of adults was caught in 2021 than 2020. In Riedstadt, there was no significant difference between the numbers of adults trapped in these two years (Table 1).

Figure 5 shows the adult abundance data, as well as the prevalence of both SBR pathogens during each calendar week, indicating a strong increase in adult abundance and the SBR infection rate between 2020 (Figure 5A) and 2021 (Figure 5B). Interestingly, in 2021, we recorded a substantial increase the number of individuals infected with both pathogens, coinciding with the elevated abundance of *P. leporinus* from calendar week 26 onwards (Figure 5B).

A closer look at the infection profiles at each site reveals interesting differences (Figure 6). In 2020, the proportion of uninfected planthoppers was 20.5–85.1% depending on the site. However, in 2021 this had fallen to 2.9–20.0%. When comparing the means of all four sites, 53.4% *P. leporinus* were singly or doubly infected in 2020 whereas this had risen to 89.4% in 2021.

In both years, and at all sites except Riedstadt in 2021, SBRars was the dominant pathogen. Interestingly, 94.3% of the planthoppers examined in Riedstadt, tested positive for infection with SBRps.

## 4. Discussion

The development of *P. leporinus* mass rearing protocols not only facilitated the life cycle studies described herein but will also allow the analysis of trait parameters that can be used to predict the future spreading and behavior of this pest. Furthermore, the mass rearing of *P. leporinus* is necessary to establish bioassays for the development of biological and environmentally friendly control methods. We developed a *P. leporinus* mass rearing method that was satisfactory in two different settings: a climate-controlled cabinet and greenhouse. Rearing in the climate-controlled cabinet required more effort than the greenhouse, but made it easier to collect adults and nymphs, and to trace them through their developmental stages. A key factor underpinning the success of rearing was the moderate and stringent watering regime and the use of GeoHumus or expanded clay to provide hiding places for oviposition and molting. The absence of an obligate diapause during the development of *P. leporinus* was in line with the field studies, where nymphs at different soil depths remained active even at low temperatures during winter.

The results of the host plant preference tests were important for several reasons. First, they showed that sugar beet is an attractive host for nymphs as well as adults, so there is no need to provide different host plant material during mass rearing. Second, they confirmed that nymphs are polyphagous, and indeed show no preference when offered a choice between sugar beet and maize. Winter wheat is the most common following crop after sugar beets, and it has been hypothesized that winter wheat might be a suitable host plant for *P. leporinus* nymphs. However, our host plant preference test did not confirm a high attractiveness of winter wheat. One reason for the lack of attractiveness of wheat could be the amount and composition of volatiles released by the single plants. For further evaluation of the attractiveness for *P. leporinus* nymphs solid-phase micro-extraction-gas chromatography-mass spectrometry could be performed [15].

The initial host plant of *P. leporinus* was the common reed *Phragmites australis*, but the circumstances surrounding the switch from reed to sugar beet are unclear [16]. It would be interesting to check for *P. leporinus* strains that are still endemic in aquatic or semi-aquatic *P. australis* and to compare them with strains that have adapted to sugar beet. Planthoppers have colonized subterranean habitats, as exemplified by the cavern-dwelling genus *Oliarus*, and might serve as a model for rapid subterranean speciation [17,18].

Nymph nutrition and movement are important factors to consider when evaluating pest control strategies based on limiting the food supply of insect pests or the use of agronomic practices that disrupt their life cycle [19]. We found that the nymphs are not restricted to sugar beet, only showing a weak preference for sugar beet over winter wheat and no preference between sugar beet and maize. The suggestion to switch from winter wheat to maize as a post-beet crop is therefore unlikely to inhibit *P. leporinus* development. In terms of movement, it was shown that *P. leporinus* nymphs develop underground and move through different soil layers based on sampling carried out in September, before sugar beet harvesting, and in May, just before adult emergence [1]. Our results from weekly sampling show that nymphs migrate to the topsoil (0–10 cm) when the temperature reaches an average of 8.6 °C, providing an opportunity to use ploughing as a means to disrupt the life cycle as part of an integrated pest management program. Sugar beet is harvested between September and November, when the soil temperature in our study areas falls from an average of 13.7 °C to 6.6 °C, so ploughing soon after harvest would seem a feasible strategy to interfere with *P. leporinus* movement. However, we also found that nymphs remained active even when the soil temperature dropped to 0 °C, similar to other fulgoromorph species such as *Hyalesthes obsoletus* larvae found on lavender [20].

We have provided the first experimental observations of *P. leporinus* movement patterns in south Hesse and Rhineland-Palatinate. SBR was first detected in sugar beet roots in Heilbronn, Baden-Württemberg, in 2010 [20], and the Palatinate has been infested with SBR since 2018 [21]. No SBR infection was detected in the Riedstadt district of Groß-Gerau during yellowing disease monitoring in 2018 and 2019 [4]. Infected sugar beet roots were first detected in the district of Groß-Gerau (south Hesse) in 2020 [4]. The significant increase in the *P. leporinus* population in south Hesse and the Palatinate between 2020 and 2021 may indicate the recent colonization of sugar beet crops in this area. Our field studies revealed that stolbur phytoplasma infection increased significantly between 2020 and 2021. To our knowledge, this is the first time a phytoplasma infection has exceeded 15% of a *P. leporinus* population, which is alarming due to the key role of phytoplasma in SBR-induced damage to sugar beet, especially in south Hesse. Further studies are needed to understand the interactions between these pathogens and their host plant and insect vector. Including phytoplasma in future monitoring activity could help to predict the mass movement of *P. leporinus* and thus prevent the spread of SBR by allowing the deployment of appropriate countermeasures. Another possibility is that the habitat is already infested the stolbur phytoplasma. *Candidatus* Phytoplasma solani is also transmitted by *H. obsoletus* in potato [22]. In 2021, stolbur phytoplasma caused tremendous damage in south Hesse and the Palatinate by reducing the marketability of potato tubers (Michael Lenz, Regierungspräsidium Hessen, personal communication). The cross-infestation of potato and sugar beet by vector insects has yet to be investigated.

The movement pattern we observed was similar to that reported in Baden-Württemberg [21] and Burgundy/Franche-Comté [1]. The emergence of adults is subject to annual fluctuations. The main flight period is calendar weeks 23–31 with a peak during week 25. The long flight period of *P. leporinus* is the main challenge when seeking control strategies. *P. leporinus* females carrying SBRars were shown to remain infectious over their entire adult lifespan of 30 days [9], whereas SBRps transmission assays by the same group were carried out using *H. obsoletus* because no infected *P. leporinus* could be found at the time of their study [23]. The two pathogens affected sugar beet plants in different ways, with the sugar content (Brix) of the roots and the mean tap root biomass as a proportion of total plant biomass affected more severely by SBRps than SBRars [24]. No data are yet available showing the effect of dual infections on sugar beet yields, but this information would facilitate the development of suitable SBR control measures.

## 5. Conclusions

We have established mass rearing techniques for *P. leporinus* allowing us to study life cycle and ecological traits that will facilitate the development of control strategies, including agronomic practices such as tilling and biological control methods. We found that *P. leporinus* nymphs do not appear to hibernate overwinter and move to the topsoil when the average temperature is ~9 °C. Host-choice experiments indicated a broad host range. Furthermore, we conducted field studies in recently infested regions of Germany to better understand the movement patterns of this insect. The results indicate that *P. leporinus* is spreading rapidly in Germany. Finally, we developed an accurate and sensitive protocol to detect the two *bona fide* SBR pathogens *Candidatus* Arsenophonus phytopathogenicus and *Candidatus* Phytoplasma solani, showing that the number of infected *P. leporinus* increased during the 2-year course of our study.

## Figures and Tables

**Figure 1 insects-13-00656-f001:**
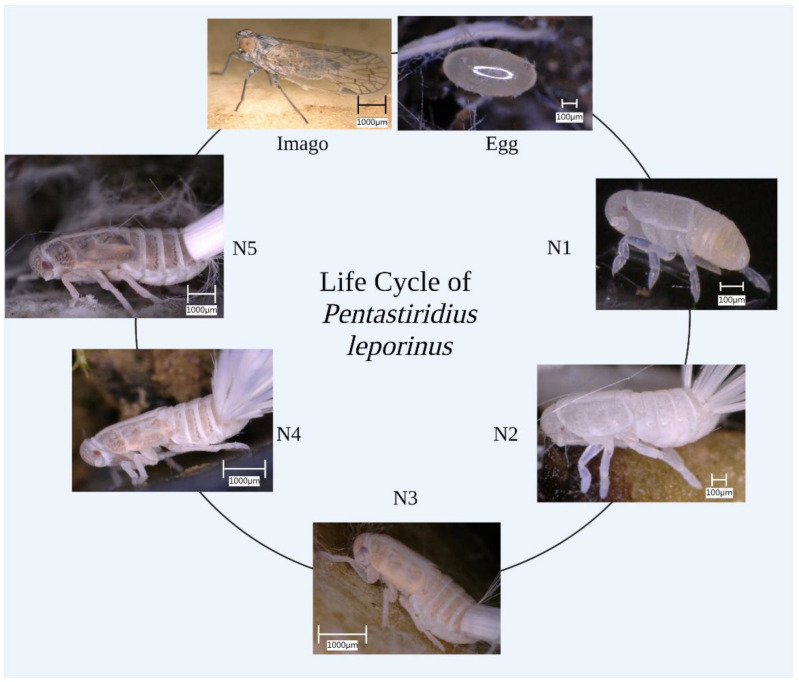
The life cycle of *Pentastiridius*
*leporinus*. Development from the egg to the adult imago proceeds through five instar nymphs (N1–N5).

**Figure 2 insects-13-00656-f002:**
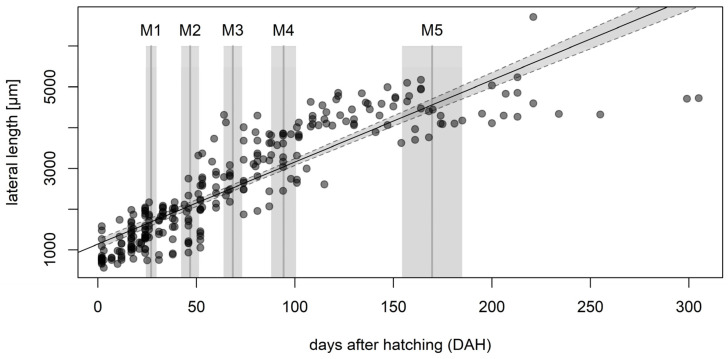
Growth rate of *Pentastiridius leporinus*. The body length (shown as lateral length in μm on the *y*-axis) was determined for *n* = 60 nymphs over a period of 34 weeks. Light gray intervals around molting time points (dark gray bars) represent ± one standard error of the mean. A linear regression yielded the function *y* = 20.6x + 1176.4, which is displayed as a solid black line with gray dashed lines indicating the 95% confidence interval.

**Figure 3 insects-13-00656-f003:**
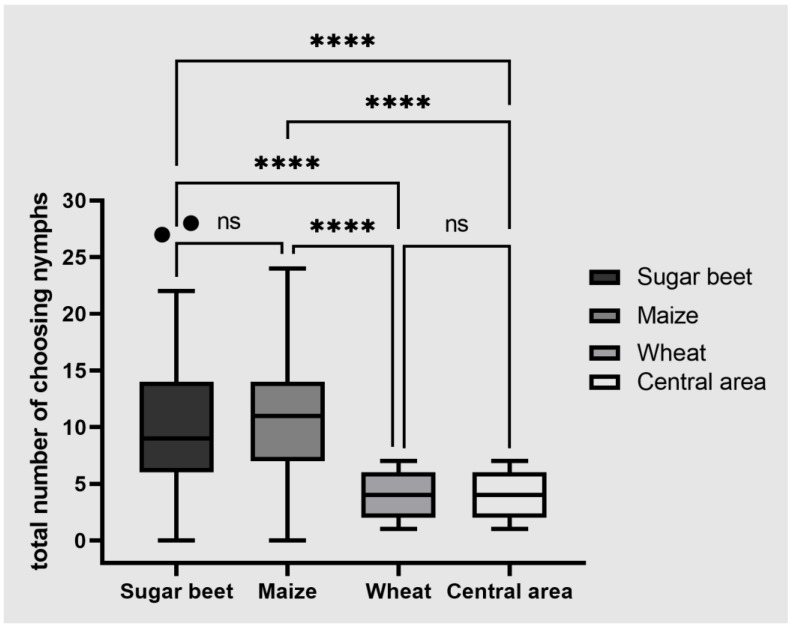
Host choice experiment in which *Pentastiridius leporinus* nymphs were placed in the central area and given a free choice of three alternative hosts. The number of nymphs found in the four areas of the test was counted after 1, 3, 6, 12, 24 and 48 h. The box plot shows the distribution of nymphs in all four areas averaged over all time points. Data were analyzed using a Friedman test. Boxes span the 25th to 75th percentiles, and the horizontal line in the box represents the median. The whiskers and outliers are plotted using Tukey’s method. Outliers are plotted as individual points. Each experiment was carried out with five sets of 30 nymphs. Statistical significance: **** *p* ≤ 0.0001; ns, not significant.

**Figure 4 insects-13-00656-f004:**
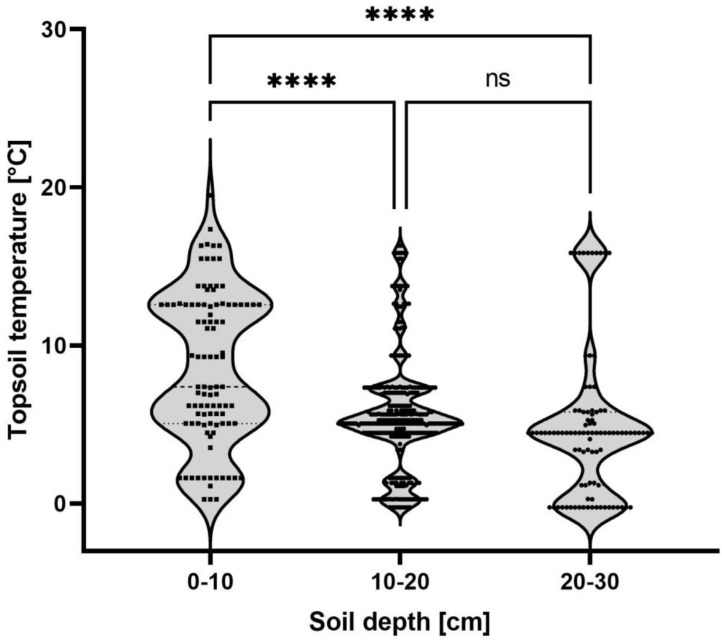
Subterranean *Pentastiridius leporinus* nymph movement. The topsoil temperature at which individual nymphs between November 2020 and March 2022 were found is plotted against the soil depth. In June, the site was changed due to the emergence of *P. leporinus* adults. Sample sizes per group: 0–10 cm (*n* = 95), 10–20 cm (*n* = 298), 20–30 cm (*n* = 95). Each data point represents one nymph. Data were analyzed by one-way ANOVA. Violin plots indicate the frequency distribution of the nymph movement; lines indicate median and quartiles. Statistical significance: **** *p* ≤ 0.0001; ns, not significant.

**Figure 5 insects-13-00656-f005:**
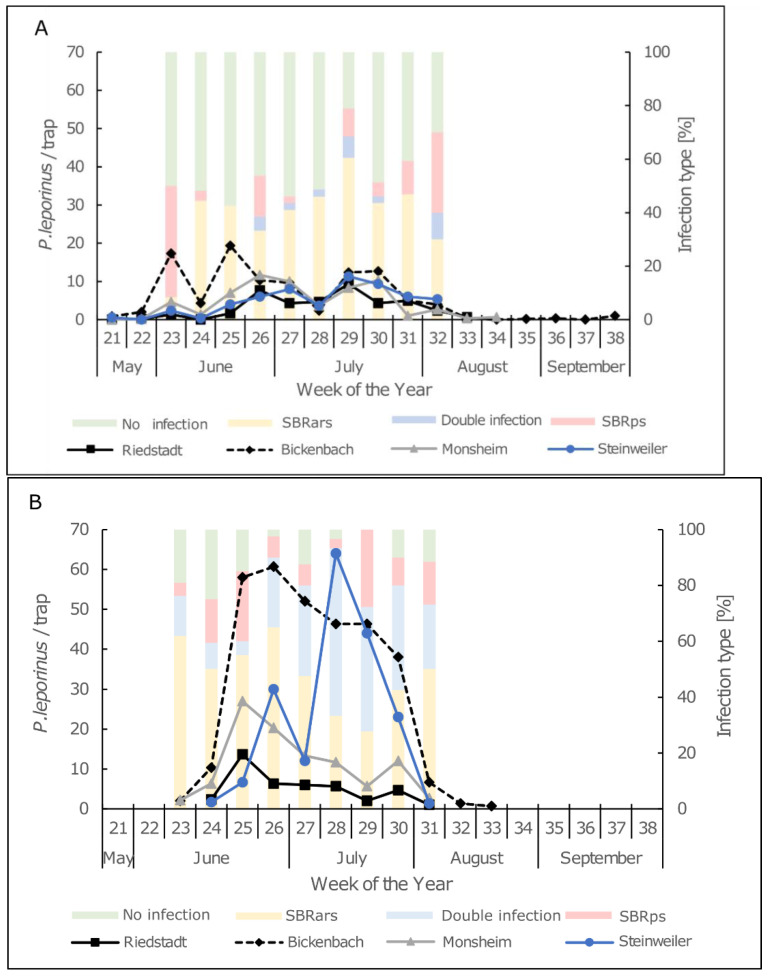
Adult abundance (lines) and infection rates (columns) of adult *P. leporinus* at four different locations in Rhineland-Palatinate and Hesse. The numbers representing adult abundance are means of three sticky traps for (**A**) 2020 and (**B**) 2021. The number of adult samples per week for the analysis of SBRars + SBRps infection rates was (**A**) 10–40 for 2020 and (**B**) 7–40 for 2021.

**Figure 6 insects-13-00656-f006:**
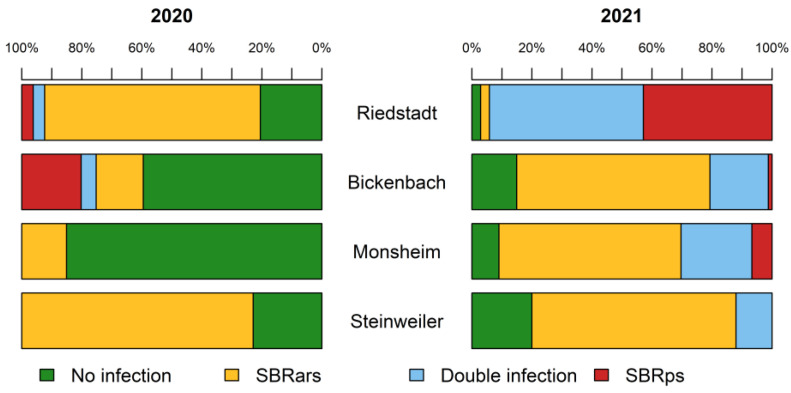
Analysis of *Pentastiridius leporinus* adults by qRT-PCR for the prevalence of the SBR pathogens *Candidatus* Arsenophonus phytopathogenicus (SBRars) and *Candidatus* Phytoplasma solani (SBRps) in 2020 (**left**) and 2021 (**right**). The number of samples tested varied per site, but 349 adults in total were tested in 2020 and 296 in 2021.

**Table 1 insects-13-00656-t001:** Abundance of *Pentastiridius*
*leporinus* in Rhineland-Palatinate (RP) and south Hesse (SH) in 2020 and 2021 during the main flight period (calendar weeks 23–31).

Site	2020	2021	Significance
Bickenbach (SH)	10.4 ± 7.1 ^1^	35.8 ± 28.7	*p* < 0.01 ^2^
Steinweiler (RP)	5.7 ± 5.4	22.6 ± 25.8	*p* < 0.05
Monsheim (RP)	6.4 ± 6.0	11.5 ± 8.5	*p* < 0.05
Riedstadt (SH)	4.3 ± 4.5	4.7 ± 4.6	*p* = 0.6

^1^ Data are means ± standard deviation. ^2^ Mann-Whitney U-test.

## Data Availability

The data presented in this study are available on request from the corresponding author.

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
