# Peer review of "Biology and Rearing of an Emerging Sugar Beet Pest: The Planthopper *Pentastiridius leporinus"

_insects, 2022, doi:10.3390/insects13070656_

Round 1

Reviewer 1 Report

The manuscript “Biology and rearing of an emerging sugar beet pest: the 2 planthopper Pentastiridius leporinus” by Behrmann et al., is a potentially interesting contribution aimed at describing the biology (life cyle), rearing method and infectivity of this insect vector of two pathogens involved in the Syndrome des Basses Richesses (SBR) of sugar beet.

However, the manuscript is not well written and is unclear to the reader in the present version. Moreover, the authors

a)    speculate far too much based on very few or weak experimental data and draw conclusions that are not supported by data

b)    address biological and evolutionary issues with little knowledge of these phenomena, so that the manuscript appears, overall, quite superficial

c)     lack scientific rigor in some points (e.g they use terms and concept such as “pathogen quantification”, “flight activity”, “nymph migration”, “nymph mobility”) uncorrectly or unclearly.

Specific comments.

Line 75 and 78: the acronyms of the pathogens chosen by the authors are counter intuitive, why not using SBRars and SBRps to remind the species/candidatus species names?

Line 152 and following paragraph. Why flight activity? What the authors mean with “flight activity”? It is rather adult population abundance. Flight activity indicate a specific behavior and yellow sticky traps, that are generally attractive because of the color, are not the proper tool to study “flight activity”. Transparent sticky traps, wind mills or other tools/methods can estimate flight activity. YST only estimate abundance of mobile forms of insects.

Line 175 and following paragraph. There is no description at all of “Quantification of SBRp and SBRst loads by qRT-PCR”. How the authors quantified the bacterial loads in the insects? Did they use standard/reference curve produced by serial dilutions of the target sequence cloned in a plasmid? Did they simply make a rough estimate based on the threshold cycle? In this latter case did they quantify the amount of total DNA that was loaded in the RT-PCR assay (very minimum requirement to produce a semi-quantitative estimation)? It seems to me that the proper title of this paragraph (also in the light of the corresponding results) is “Detection of SBRp and SBRst by qRT-PCR”

Line 207. When you say that a species is monovoltine, then using the term “second generation of adults” is misleading

Lines 212-214. Totally obscure to me. “reproduced over several life cycles”?? (one per year or several per year? Or whatever else?). “In the climate chamber, offspring developed from the eggs laid by the parental generation (P0) and the next generation (F1)”? Do the authors mean that they obtained and followed two successive generations under climate chamber conditions? Unclear  

Line 237. Central area?? There is no mention in mat&met of this “central area”. Only in caption to figure 3 the authors mention this central area, but they should better describe the experiment in the proper section, mat&met.

Line 248 and following paragraph. It is not clear how temperature-dependent mobility was estimated. From the description it seems that they simply count the number of nymphs at different soil depths characterized by different soil temperatures. How this describes nymph mobility? Maybe the authors actually described nymph mobility, but then the description of the experiment is not clear.

Line 266 and following paragraph. There is inconsistency with mat&met where the authors state “quantification of SBR…” and these results, where they only provide prevalence data

Figure 5. Adult abundance is represented by site, while apparently infection rates are pooled together over all the sites. Isn't it? If so, this is not very informative

Line 316. “optimal humidity was essential for the development of nymphs”: the reader would like to know what is the optimal humidity! I did not see any hint in the paper.

Lines 324-330. Purely speculative, the authors do not provide data to support this hypothesis, they simply comment on other papers. This is out of the frame of the work done. Delete.

Lines 349-352. too speculative! You need time series to conclude on that.

Lines 362-368. Please skip this line. They are purely speculative, beyond the scope of the paper and based on poor knowledge of these complex interactions. There is much more than the two cited papers (see below the papers, mainly from the Hogenhout lab). Still my suggestion is to skip this part of the discussion.

Line 372. Replace the obscure citation with the following: Mitrović, M., Jakovljević, M., Jović, J., Krstić, O., Kosovac, A., Trivellone, V., ... & Cvrković, T. (2016). ‘Candidatus phytoplasma solani’genotypes associated with potato stolbur in Serbia and the role of Hyalesthes obsoletus and Reptalus panzeri (hemiptera, cixiidae) as natural vectors. European Journal of Plant Pathology, 144(3), 619-630.

Al-Subhi, A. M., Al-Sadi, A. M., Al-Yahyai, R. A., Chen, Y., Mathers, T., Orlovskis, Z., ... & Hogenhout, S. A. (2021). Witches’ broom disease of lime contributes to phytoplasma epidemics and attracts insect vectors. Plant Disease105(9), 2637-2648.

Orlovskis, Z., & Hogenhout, S. A. (2016). A bacterial parasite effector mediates insect vector attraction in host plants independently of developmental changes. Frontiers in plant science7, 885.

MacLean, A. M., Orlovskis, Z., Kowitwanich, K., Zdziarska, A. M., Angenent, G. C., Immink, R. G., & Hogenhout, S. A. (2014). Phytoplasma effector SAP54 hijacks plant reproduction by degrading MADS-box proteins and promotes insect colonization in a RAD23-dependent manner. PLoS biology12(4), e1001835.

Sugio, A., Kingdom, H. N., MacLean, A. M., Grieve, V. M., & Hogenhout, S. A. (2011). Phytoplasma protein effector SAP11 enhances insect vector reproduction by manipulating plant development and defense hormone biosynthesis. Proceedings of the National Academy of Sciences108(48), E1254-E1263.

Sugio, A., MacLean, A. M., Kingdom, H. N., Grieve, V. M., Manimekalai, R., & Hogenhout, S. A. (2011). Diverse targets of phytoplasma effectors: from plant development to defense against insects. Annual review of phytopathology49(1), 175-195.

Hogenhout, S. A., Oshima, K., AMMAR, E. D., Kakizawa, S., Kingdom, H. N., & Namba, S. (2008). Phytoplasmas: bacteria that manipulate plants and insects. Molecular plant pathology9(4), 403-423.

Pradit, N., Mescher, M. C., De Moraes, C. M., & Rodriguez-Saona, C. (2020). Phytoplasma infection of cranberry affects development and oviposition, but not host-plant selection, of the insect vector Limotettix vaccinii. Journal of chemical ecology46(8), 722-734.

Author Response

Reviewer 1:

Comments and Suggestions for Authors

The manuscript “Biology and rearing of an emerging sugar beet pest: the 2 planthopper Pentastiridius leporinus” by Behrmann et al., is a potentially interesting contribution aimed at describing the biology (life cyle), rearing method and infectivity of this insect vector of two pathogens involved in the Syndrome des Basses Richesses (SBR) of sugar beet.

However, the manuscript is not well written and is unclear to the reader in the present version. Moreover, the authors

  1. a) speculate far too much based on very few or weak experimental data and draw conclusions that are not supported by data
  2. b) address biological and evolutionary issues with little knowledge of these phenomena, so that the manuscript appears, overall, quite superficial
  3. c) lack scientific rigor in some points (e.g they use terms and concept such as “pathogen quantification”, “flight activity”, “nymph migration”, “nymph mobility”) uncorrectly or unclearly.

We thank reviewer 1 for his input and implemented his suggestions and recommendations to increase the quality of the manuscript. We took particular care on point c) to be more precise in the terms pathogen quantification”, “flight activity”, “nymph migration”, “nymph mobility”.

Specific comments.

Line 75 and 78: the acronyms of the pathogens chosen by the authors are counter intuitive, why not using SBRars and SBRps to remind the species/candidatus species names?

We thank the reviewer for his suggestion and changed the names in the manuscript and the figures accordingly.

 Line 152 and following paragraph. Why flight activity? What the authors mean with “flight activity”? It is rather adult population abundance. Flight activity indicate a specific behavior and yellow sticky traps, that are generally attractive because of the color, are not the proper tool to study “flight activity”. Transparent sticky traps, wind mills or other tools/methods can estimate flight activity. YST only estimate abundance of mobile forms of insects.

We followed the suggestion of reviewer 1 and changed flight activity to abundance of adults.

Line 175 and following paragraph. There is no description at all of “Quantification of SBRp and SBRst loads by qRT-PCR”. How the authors quantified the bacterial loads in the insects? Did they use standard/reference curve produced by serial dilutions of the target sequence cloned in a plasmid? Did they simply make a rough estimate based on the threshold cycle? In this latter case did they quantify the amount of total DNA that was loaded in the RT-PCR assay (very minimum requirement to produce a semi-quantitative estimation)? It seems to me that the proper title of this paragraph (also in the light of the corresponding results) is “Detection of SBRp and SBRst by qRT-PCR”

We used a standard curve and included the missing information in the materials & methods section. However, we defined a threshold cycle of Ct 35, corresponding to 10 and 3 copies for SBRars and SBRps, respectively; below this threshold, the samples were considered positive for the tested pathogens.  As suggested, we changed the title to “Detection of SBRp and SBRst by qRT-PCR” and added further information.

Line 207. When you say that a species is monovoltine, then using the term “second generation of adults” is misleading

The univoltine / monovoltine statement we did was for field conditions. The results are describing lab and greenhouse conditions. We added “after three months under climate controlled conditions.” to clarify this point.

Lines 212-214. Totally obscure to me. “reproduced over several life cycles”?? (one per year or several per year? Or whatever else?). “In the climate chamber, offspring developed from the eggs laid by the parental generation (P0) and the next generation (F1)”? Do the authors mean that they obtained and followed two successive generations under climate chamber conditions? Unclear 

Thank you for the suggestion, we changed the paragraph to “From the initial P. leporinus population (collected in the field in summer 2021) we studied two successive generations under climate chamber conditions (Figure 1).” to increase the clarity of the procedures.

Line 237. Central area?? There is no mention in mat&met of this “central area”. Only in caption to figure 3 the authors mention this central area, but they should better describe the experiment in the proper section, mat&met.

Thank you for the suggestion, we changed the paragraph in materials and methods accordingly.

Line 248 and following paragraph. It is not clear how temperature-dependent mobility was estimated. From the description it seems that they simply count the number of nymphs at different soil depths characterized by different soil temperatures. How this describes nymph mobility? Maybe the authors actually described nymph mobility, but then the description of the experiment is not clear.

We recorded the abundance of nymphs at different soil depth as a function of soil temperature in 10 cm depth, and changed the paragraph for better understanding.

Line 266 and following paragraph. There is inconsistency with mat&met where the authors state “quantification of SBR…” and these results, where they only provide prevalence data

As previously suggested by reviewer 1, we changed the title to “Abundance of adults of P. leporinus and prevalence of SBR pathogens”.

Figure 5. Adult abundance is represented by site, while apparently infection rates are pooled together over all the sites. Isn't it? If so, this is not very informative

We kindly disagree to this point raised by reviewer 1. The adult planthopper abundance together with the prevalence rate is providing a first hint in the diffusion of SBR in two successive years; especially the increase of SBRps and the double infections are relevant to sugar beet farming. In addition, we have shown in Figure 6 the infection rates per site for two consecutive years.

Line 316. “optimal humidity was essential for the development of nymphs”: the reader would like to know what is the optimal humidity! I did not see any hint in the paper.

We agree that we did not present this in the results section and removed the sentence.

Lines 324-330. Purely speculative, the authors do not provide data to support this hypothesis, they simply comment on other papers. This is out of the frame of the work done. Delete.

We are referring to previous publications and are setting this in the context of our host choice experiments, suggesting some future studies to this interesting question.

Lines 349-352. too speculative! You need time series to conclude on that.

We started with our statement, followed by the argumentation. We have now moved the statement sentence after the line of argumentation to support our hypothesis, please note that we do acknowledge the (so far) hypothetical nature by using “may indicate”.

Lines 362-368. Please skip this line. They are purely speculative, beyond the scope of the paper and based on poor knowledge of these complex interactions. There is much more than the two cited papers (see below the papers, mainly from the Hogenhout lab). Still my suggestion is to skip this part of the discussion.

We agree that currently we have no strong support for this interesting idea. We will therefore follow the suggestion of reviewer 1 and skip this part of the discussion.

Line 372. Replace the obscure citation with the following: Mitrović, M., Jakovljević, M., Jović, J., Krstić, O., Kosovac, A., Trivellone, V., ... & Cvrković, T. (2016). ‘Candidatus phytoplasma solani’genotypes associated with potato stolbur in Serbia and the role of Hyalesthes obsoletus and Reptalus panzeri (hemiptera, cixiidae) as natural vectors. European Journal of Plant Pathology, 144(3), 619-630.

We changed the reference accordingly

Reviewer 2 Report

The manuscript of Behrmann et al. described the biology and rearing of planthopper Pentastiridius leporinus. They found that P. leporinus nymphs do not appear to hibernate or overwinter and P. leporinus adults are most active during June and July. Furthermore, they developed and tested a protocol to detect SBR pathogens Candidatus Arsenophonus phytopathogenicus and Candidatus Phytoplasma solani. Data generated from this study have a high significance in terms of both pathogen and vector management programs. The presentation of methods and results was clear and well described. In addition, the details in the introduction and discussion section are also well connected with the methods and results. Based on my comments, the manuscript needs only minor edits (comments and edits in the attached file). 

Author Response

Reviewer 2

The manuscript of Behrmann et al. described the biology and rearing of planthopper Pentastiridius leporinus. They found that P. leporinus nymphs do not appear to hibernate or overwinter and P. leporinus adults are most active during June and July. Furthermore, they developed and tested a protocol to detect SBR pathogens Candidatus Arsenophonus phytopathogenicus and Candidatus Phytoplasma solani. Data generated from this study have a high significance in terms of both pathogen and vector management programs. The presentation of methods and results was clear and well described. In addition, the details in the introduction and discussion section are also well connected with the methods and results. Based on my comments, the manuscript needs only minor edits (comments and edits in the attached file). 

We thank reviewer 2 for his positive comments.

Line 113. “a bit confusing as you collected adults”

We changed the sentence accordingly.

Line 144. “briefly describe the composition of nutrient solution”

We added the composition of nutrient solution as requested in the Material and Methods section.

Line 238 “Since single plant from each host was used, there might be leaf area effects on amount of volatile released by the plants. Thus affecting host preference. Please include this in the discussion.”

We added a paragraph in line 324 to address the point of reviewer 2.

Line 268. “marking 2022 individuals in year 2021.”

We know this looks wrong and like a typo, but it is the right number we found.

Line 338. “something is missing here”

We thank the reviewer for his comment and completed the paragraph accordingly.

Line 367, “vector born plant pathogens can also provide fitness benefits to their vectors by alerting plant biochemistry or immune responses. Thereby assisting in vector population growth.”

Following reviewer 1`s comments we deleted this section in the discussion.

Line 371 “Correcting “is already infested “with” the stolbur phytoplasma”

We followed the suggestion of reviewer 2 and changed the sentence.

Round 2

Reviewer 1 Report

The manuscript has been substantially improved and is now more clear and rigorous/precise. 

Still I have a remark on the paragraph of Subterranean P. leporinus nymphs migration. First of all I recommend the authors to replace the term migration with movement. I was not happy with the term mobility, but I disagree even more with the term "migration" that, namely in entomology, has a precise meaning, that does not apply to this case. So, please use the sentence "Subterranean P. leporinus nymphs movement". Moreover, I do not like very much Fig. 4. In its present form fig. 4 plots something (which is not indicated) against soil Temp and soil depth. I know that you plotted number of nymphs, but this is not stated in the graph. I wonder if it is interesting to point out differences in the nymph abundance among the different soil layers or, better, to show nymph distribution in relation to soil temperature/depth with a dispersal point graph and a fitting of the nymph distribution. Moreover, the Y axis is named "soil temp" but it is rather the "top soil temperature", am I right? I suggest the use of a different type of graph, but in the end if the authors prefer to keep this one they can keep it, provided that it is fully self-explained and correct.

The link at lines 388-9 does not work, please amend it.

Author Response

Reviewer 1

The manuscript has been substantially improved and is now more clear and rigorous/precise. 

We thank reviewer 1 for his valuable input to increase the quality of the manuscript.

Still I have a remark on the paragraph of Subterranean P. leporinus nymphs migration. First of all I recommend the authors to replace the term migration with movement. I was not happy with the term mobility, but I disagree even more with the term "migration" that, namely in entomology, has a precise meaning, that does not apply to this case. So, please use the sentence "Subterranean P. leporinus nymphs movement".

We changed as requested by reviewer 1 the term migration to movement throughout the manuscript.

Moreover, I do not like very much Fig. 4. In its present form fig. 4 plots something (which is not indicated) against soil Temp and soil depth. I know that you plotted number of nymphs, but this is not stated in the graph. I wonder if it is interesting to point out differences in the nymph abundance among the different soil layers or, better, to show nymph distribution in relation to soil temperature/depth with a dispersal point graph and a fitting of the nymph distribution. Moreover, the Y axis is named "soil temp" but it is rather the "top soil temperature", am I right? I suggest the use of a different type of graph, but in the end if the authors prefer to keep this one they can keep it, provided that it is fully self-explained and correct.

We thank reviewer 1 for his input. We have now enhanced Fig. 4 as requested and changed the data analysis from events to single nymphs at different soil layers. We plotted Topsoil temperature against soil depth in a violin plot, each dot representing one nymph. We hope that we increased the clarity of presentation and provide a better insight in nymph distribution.

The link at lines 388-9 does not work, please amend it.

We apologize for our mistake; the link was a placeholder for a reference. We have now changed the link  reference [15], as intended.
